# Corona Resistance Mechanism of Nano-Modified Polyimide

**DOI:** 10.3390/polym14245469

**Published:** 2022-12-14

**Authors:** Hao Chen, Lin Li, Wei Zhao, Xiao-Rui Zhang, Ling Weng

**Affiliations:** Department of Polymer Materials and Engineering, College of Material Science and Chemical Engineering, Harbin University of Science and Technology, 4 Linyuan Road, Harbin 150040, China

**Keywords:** polyimide, nano-SiO_2_ particles, electrical aging threshold, corona resistance, dielectric properties

## Abstract

In this paper, the effect of field strength on the corona-resistant lifespan of a composite film and the effect of doping on the dielectric properties of the composite film were studied. The method for predicting corona-resistant lifespan under working electric field strength is discussed. Scanning electron microscopy (SEM) was used to characterize the morphology and the structure of the composite film near the breakdown point after corona formation. Fourier transform infrared spectroscopy (FT-IR) was used to characterize the imidiated film, and a conductivity current test was used to calculate the electrical aging threshold of the film. The results showed that the introduction of nano-SiO_2_ particles could greatly improve the corona-resistant lifespan of the material. At 155 °C, when the applied external electric field strength was extrapolated to 20 kV/mm, the corona-resistant lifespan of the PI/nano-SiO_2_ three-layer composite film with 10 wt% nano-particle doping was 7472.61 h.

## 1. Introduction

In recent years, the development of high-speed railway systems has achieved unprecedented results. The variable frequency speed-regulating traction motor is one of the critical components of high-speed EMU power equipment and its performance is key to ensuring stable and reliable operation of the train. In the variable frequency speed-regulation traction system, due to the motor-generated overvoltage [1,2,3] coupled with the uneven distribution of pulse voltage on the winding group, corona discharge occurs between winding turns, so that the insulation system of the motor is destroyed due to premature breakdown [4,5,6]. Studies have shown that corona discharge in the inter-turn insulation of the motor is the root cause of motor insulation failure. Therefore, improving the corona resistance of the inter-turn insulation of the variable frequency motor can significantly slow the insulation aging process of the motor.

Polyimide (PI) is a kind of polymer with an imide ring structure in its main chain. Electrical properties, radiation resistance, chemical resistance, good electrical insulation, and chemical stability of polyimide are particularly prominent. It has been widely used in various fields [7,8,9,10,11,12,13,14,15,16,17,18,19,20]. Polyimide has become the key to development of engineering plastics in many countries. Polyimide films have been widely used as the most important insulation material for variable frequency traction motors in motor slot insulation, inter-turn insulation, and ground insulation. Researchers all over the world are focusing on the research of corona-resistant insulation materials. Experiments show that [21,22,23,24,25,26,27,28,29,30,31,32] inorganic oxides can improve the properties of the matrix to a certain extent, and adding a certain amount of inorganic nano-oxide to a polyimide matrix can greatly improve the corona resistance of polymer insulation materials, thus meeting the operation requirements of the variable frequency motor and ensuring its service life. At the same time, the urgent problem to be solved is clarifying the mechanism of PI film corona resistance as soon as possible, which can provide a feasible technical route for industrial production. Therefore, it is of great practical significance to improve the service life of low-voltage electrical equipment and the insulation properties of materials, as well as to study polyimide nanocomposites with good corona resistance.

Relevant literature [33] has inferred the corona-resistant lifespan at working field strength (20 kV/mm) through the relationship between field strength and corona resistance time, but does not consider the relationship between electrical aging threshold and corona-resistant lifespan, the factors considered more comprehensively in this paper. Pure PI films and polyimide films with nano-SiO_2_ doping amounts of 5 wt%, 10 wt%, and 15 wt%, respectively, were taken as the research objects. According to the corona resistance time and conductivity current test at high temperature and high field strength, the electrical aging threshold was obtained, and then the corona-resistant lifespan at normal temperature and working field strength was inferred from the electrical aging threshold. The results are relatively realistic.

Based on the structural characterization and dielectric properties of nano-SiO_2_ modified polyimide films, this paper discusses the corona resistance of films and the comparative method to infer the corona-resistant lifespan based on the electrical aging threshold.

## 2. Materials and Methods

### 2.1. Main Experimental Materials and Experimental Apparatus

4,4′-diamino-diphenyl ether (ODA): Mudanjiang Insulation Material Factory (Mudanjiang, China), industrial product, the premise of use is pure; N, N′-dimethyl acetamide (DMAc): Mudanjiang Insulation Material Factory, industrial products, the premise of use is pure; phthalic anhydride (PMDA): Shanghai Reagent Factory No. 3 (Shanghai, China), analytically pure; ethyl orthosilicate (TEOS): Shenyang Reagent Factory No. 3 (Shenyang, China), chemically pure; methyl triethoxysilane (MTEOS): Tianjin Chemical Reagent Factory (Tianjin, China), chemically pure; toluene: Harbin Reagent Factory (Harbin, China), chemically pure.

Three-necked bottle: Renhe Quartz Glass Instrument Factory (Jinzhou, China); EST122-picoammeter: Beijing Huace Experimental Instrument Company (Beijing, China); HV-103P2 DC high voltage power supply: Dongguan Napu Electric Technology Company (Dongguan, China); SU8020 scanning electron microscopy: Hatachi Company (Ibaraki, Japan); Nicolet iS50 Fourier transform infrared spectrometer: Thermo Fisher Scientific (Waltham, MA, America); Alpha-A broadband dielectric spectrum analyzer: Novocontrol Technologies (Montabaur, Germany).

### 2.2. Preparation of Nano-SiO_2_ Dispersion

In this experiment, the nano-SiO_2_ dispersion was prepared in two methods. The first method is combining micro-emulsion: Anhydrous ethanol, distilled water, tetraethyl orthosilicate, and methyl triethoxysilane were added into a 250 mL three-necked bottle according to the prescribed ratio, the mixture was stirred quickly and maintained at 80 °C for 4 h, and then the temperature was raised to 120 °C. The organic nano-sol was prepared by removing the water and alcohol in the solution by distillation. The second method is supercritical: the nano-glue solution was transferred to a self-made super-critical reaction vessel, the temperature was increased for 2 h of super-critical reaction. Finally, the modified inorganic nano-SiO_2_ sol was obtained. The prepared nano-SiO_2_ sol was transferred to a beaker and aged for one week at room temperature under dry, sealed conditions. The purpose of aging was to maintain slow polymerization between the colloidal particles in order to form a three-dimensional network structure of the sol with stable size and SiO_2_ properties, and ensure smooth progression of subsequent work.

### 2.3. Preparation of Nano-Hybrid Composite Films

A certain amount of ODA was weighed in a clean three-necked bottle, then DMAc was added at a solid content of 15% and stirred evenly to dissolve the ODA completely. Nano-dispersion liquid of a certain amount was added according to the required doping amount and stirred continuously until uniform. An equimolar amount of PMDA was added to the three-necked bottle in small amounts several times. When the amount of PMDA was added to the equimolar quantity, the viscosity of the system was suddenly increased, and an apparent phenomenon of climbing rods appeared under the action of stirring. Stirring was continued for 24 h at room temperature and the PAA glue doped with nano-SiO_2_ was finally obtained. An 800-mesh filter screen was used to perform suction filtration to remove dust particles and mechanical impurities in the glue solution. Air bubbles in the glue solution were removed by vacuuming and leaving it for filming.

The prepared PAA glue was poured into the impregnation tank with a self-made laboratory mold (with special gaps); the single-layer film was removed vertically from the glue after stirring and placed into the glue through the gap of the self-made mold, and then heated it 160 °C for about 2 min for the purpose of removing the solvent on the surfaces of the polyimide films. Then the temperature was increased to 380 °C, the three layers of nano-modified polyimide film with a thickness of 25 μm were taken out, and the middle layer was a pure polyimide film. The hybrid film was cut to the required size to be tested. Figure 1 is a schematic diagram of the procedure used to prepare the composite films.

### 2.4. Testing and Characterization

Corona resistance tests were performed on PI/SiO_2_ films with different doping amounts using a corona resistance tester assembled according to IEC 60343 standards; conductance current tests were performed with EST122-picoammeter and HV-103P2 DC high voltage power supply; the SU8020 scanning electron microscopy (SEM)was used to characterize the morphology of the breakdown point of the film and surroundings after corona formation. The microstructures were characterized by using a Thermo Scientific Nicolet iS50 Fourier transform infrared spectrometer with an attenuated total reflection (ATR) accessory; the dielectric properties of the composite films were tested with an Alpha-A broadband dielectric spectrum analyzer.

## 3. Results and Discussion

### 3.1. Morphology of the Composite Films

Scanning electron microscopy (SEM) was used to observe the morphology of the film section and the surface of the film after corona breakdown erosion. Figure 2 shows the cross-sectional topography of a three-layer PI/nano-SiO_2_ film with the doping amount of 5 wt%. The middle layer was the pure PI film with a thickness of about 13 μm, and the PI film on both sides was doped with nano-SiO_2_ with a thickness of about 6 μm. The average size of nano-SiO_2_ particles in the doped layer was about 20 nm; the distribution in the doped layer was relatively uniform without obvious agglomeration phenomenon.

Figure 3 shows the TEM diagram with doping amount of 5 wt% PI/nano-SiO_2_ three-layer composite film enlarged 80,000 times. In the diagram, the dark area is nano-SiO_2_ inorganic phase, the lighter area is PI matrix organic phase, and the bright area is PI matrix damage caused by the high degree of hardness of inorganic nanoparticles when TEM sample slices were prepared. It can be seen from Figure 3 that the inorganic/organic interface was complete with no air gaps between the nano-SiO_2_ particles and the PI matrix. The presence of voids around some of the nanoparticles in the SEM image of Figure 2 was also due to the extraction of some of the nanoparticles during the preparation of the SEM sample. Due to the poor conductivity of the insulation material system, if the carbon spraying treatment could not effectively improve the conductivity, the clarity of the SEM and TEM tests was not effectively improved, even under magnification.

Figure 4 shows the SEM surface topography near the breakdown point of the PI/nano-SiO_2_ three-layer composite film with a doping amount of 5 wt% after corona breakdown. After breakdown, the matrix of the polymer was eroded, the surface became rough compared with the initially smooth film, and the breakdown point was carbonized. After corona formation, a precipitate of nanoparticles gathered on the surface of the film. The reason is that under the action of the strong electric field generated by corona discharge, the polymer chain of the polyimide matrix was impacted, causing the polymer chain to break and melt. At the same time, the precipitation of nanoparticles hindered the application of a strong electric field. Therefore, the corona resistance of the film was improved.

### 3.2. Infrared Spectroscopy Characterization of Composite Films

Infrared spectroscopy can effectively reflect the structure information of material surface depth from several hundred nanometers to several microns. The characterization of imidized polyimide films can reflect whether the imidization of polyimide films is complete. Figure 5 shows the comparison of the infrared spectra of the pure PI film and the PI/nano-SiO_2_ three-layer composite film with a doping amount of 10 wt%. It can be seen from this figure that C=O asymmetry appeared in symmetrical stretching vibration (vibration coupling) at 1775 cm^−1^ and 1715 cm^−1^, respectively. These peaks correspond to imide. No vibration peaks of -COOH and -OH appeared at >2000 cm^−1^, indicating that the imidization of the film was completed.

### 3.3. Mechanical Property Test

As can be seen from Figure 6, with the increase of nano-SiO_2_ doping amount, the elongation of nano-SiO_2_/PI three-layer composite film showed a downward trend, while the tensile strength of the film showed a downward trend, except when the nano-SiO_2_ doping amount was 5 wt%. This is because with the increase in the amount of nano-SiO_2_ doping, the average distance between the nanoparticles decreased, and the polymer matrix could not be well connected, which will form defects. Under the action of tensile force, the defect position was prone to stress concentration, which led to rapid fracture of the material. When the doping amount of nano-SiO_2_ was 5 wt%, the breaking strength of the nano-SiO_2_/PI three-layer composite film was higher than that of pure PI film, and its elongation was less than that of pure PI film. This can be explained as a small amount of nano-SiO_2_ interacting with PI molecules, which enhanced the rigidity of the PI molecular chain and provided a certain free volume, which increased the tensile strength of the film. In addition, it can be seen from Figure 6 that although the tensile strength and elongation at breakage of the film had decreased to a certain extent, the range was not large, and the actual use was not affected. The main reason for this result is that the interlayer of the three-layer composite film was a commercial film, which has good mechanical properties. During the stretching process, it could have a considerable reinforcing effect on the doped layer. Therefore, the PI molecules in the doped layer could be fully oriented during stretching, so that the mechanical properties of the nano-SiO_2_/PI three-layer composite film were maintained at a high level.

### 3.4. Conductivity Current Test

Figure 7 shows the relationship between the conductance current and the field strength of the pure PI film and the PI/SiO_2_ three-layer composite film with doping amounts of 5 wt%, 10 wt%, and 15 wt%. The electrical aging threshold is reflected by the intersection of two segments. The properties of the material were reflected by the electrical aging threshold, that is, the minimum field strength at which the material began to age. Before the electrical aging threshold, at a low electric field, the current–voltage characteristics of the dielectric conformed to Ohm’s law. When the material’s field strength was lower than the electrical aging threshold, halo charge would not occur. With the increase of the applied voltage, the carriers injected into the material were also increased, and a large amount of space charge accumulated, which made the current change from the ohmic region to the current-limiting region. The transition voltage was the electrical aging threshold at this time. Table 1 shows the electrical aging threshold calculated from the data in Figure 7.

In the conductance current density curve, the electrical aging threshold of the film can be calculated from the turning point of the ohmic conductance region and the space-charge-limited current region. The electrical aging threshold is reflected by the electric field strength at which the material begins to accumulate a large amount of charge. This means that under the condition of injecting the same charge, the doped PI films accumulated space charge after injection of more carriers. From Table 1 we can see that the electrical aging threshold of the PI film doped with nano-SiO_2_ was significantly larger than that of the pure PI film. That is, the space-charge-limited current region was reduced, and the trap carrier density was increased. The reasons may be: (1) due to the addition of nanoparticles, the migration rate of carriers was reduced in the film; (2) the addition of nano-SiO_2_ produced more traps. Compared with pure PI film, when the amounts of injected electrons are equal, there will be many electrons falling into the traps formed by the nanoparticles, reducing the effect on the conductance current.

### 3.5. Corona Resistance Test

Figure 8 shows the relationship between the corona-resistant lifespan (L) and the applied field strength (E) of pure PI films and PI/nano-SiO_2_ films with doping amounts of 5 wt%, 10 wt%, and 15 wt%, respectively, at 155 °C. As can be seen from Figure 8, the nano-SiO_2_ particles were added to the polymer matrix as an inorganic phase and the corona resistance time of the composite film was greatly improved. This shows that the resistance to the electricity of the composite film was enhanced by inorganic particles, to a certain extent. When a strong electric field was applied to the pure PI film, the air on the surface of the film was first ionized, and the generated positive and negative ions would erode the surface of the film. With the continuous action of the applied electric field and the increase of time, the carriers inside the film accumulated. When the internal carriers accumulated to a certain extent, a conductive path would be formed, thus causing the pure PI film to break down. With the addition of nano-SiO_2_ particles, a trap (defect) structure would be formed inside the film. As more SiO_2_ particles were added, more trap structures would be generated. This structure could capture the carriers generated by ionization and reduce the probability of collisions with other carriers, which can reduce the generation of more charge. As the applied field strength decreased, the accumulation rate of charge slowed down and the rate of breakdown would also decrease.

It can be seen that the incorporation of nano-SiO_2_ particles obviously improved the corona resistance of the PI film. The reason is that the internal structure of the film was changed by nano-SiO_2_ particles. At the same time, nano-SiO_2_ particles inhibited the corrosion of the film by corona, the bond energy of the Si-O bond was high, and the thermal stability was good. Therefore, the corona resistance of the material could be improved.

It also can be seen from Figure 8 that the lnL-lnE curves of some samples showed obvious inflection points near the electrical aging threshold field strength, which indicates that the corona aging mechanism was different before and after the electrical aging threshold. Therefore, when predicting the corona-resistant lifespan of the working voltage materials, the same or similar corona-aging characteristic area should be selected as a reference. For example, the electrical aging threshold of 15 wt% PI/SiO_2_ three-layer composite film was 53.02 kV/mm at 155 °C. Therefore, the corona resistance test data below 53.02 kV/mm at 155 °C should be selected as the basis for estimating the corona-resistant lifespan of 20 kV/mm. The corona-resistant lifespan of 15 wt% PI/SiO_2_ three-layer composite film was extrapolated at 155 °C and 20 kV/mm for 7472.61 h. The electrical aging thresholds of pure PI film and 5 wt% PI/SiO_2_ three-layer composite film were relatively low, at 30.99 kV/mm and 32.78 kV/mm, respectively. The field strength was not tested in this experiment, so the lnL-lnE curve does not appear as an inflection point; obviously, the extrapolated corona-resistant lifespan of 20 kV/mm can only be used as a reference. Table 2 shows the extrapolated corona-resistant lifespan of each sample at 155 °C and 20 kV/mm.

### 3.6. Dielectric Properties of PI/SiO_2_ Composite Films

Figure 9 shows the relationship of dielectric constant (*ε*) between the pure PI film and PI/SiO_2_ three-layer composite films with different doping amounts as a function of frequency (*f*). It can be seen from the figure that the dielectric constant decreased with the increase of frequency; the dielectric constant decreased rapidly at high frequencies because at low frequencies, the polarization of the interface meter played a significant role in the dielectric constant. As the frequency increased, the speed of interface polarization was less than the speed of frequency change, resulting in a rapid decrease in the dielectric constant. With the increase of the doping amount, the dielectric constant of the three-layer hybrid films frequency increased concurrently. This may be because the dielectric constant of the SiO_2_ nanoparticles themselves was relatively large; the combination between the particles and the PI matrix increased the polarization mode inside the film, making the dielectric constant of the doped films higher than that of the pure PI films.

Figure 10 and Figure 11 show the dielectric loss (tan *δ*) and the conductivity (*σ*) of pure PI film and different doping amounts of PI/SiO_2_ three-layer composite film with frequency (*f*). It can be seen from the figure that the dielectric loss of all samples was kept at a relatively small value in the measured frequency range. The dielectric loss of pure PI film was the smallest. With the increase in SiO_2_ composition, the corresponding dielectric loss and electrical conductivity increased accordingly. This is because the electrical conductivity of the film was affected by the number of internal charges and the migration rate of internal electrons. The addition of nano-SiO_2_ particles increased the probability of space charge generation and shortened the distance of the charge, which increased the conductivity of the films. Simultaneously, the dielectric loss was affected, resulting in an increase in dielectric loss.

## 4. Conclusions

This study mainly examined the PI/nano-SiO_2_ three-layer composite corona-resistant film, and the conclusions are as follows:(1)The addition of inorganic nano-components improves the density of thermally excited carriers of the material. It increases the density of trap carriers by the improvement of the threshold of the electrically aging material. The addition of nanoparticles can improve the corona resistance and thermal conductivity of the film, reduce the erosion of the film by thermionic bombing, and make the corona resistance time of the film increase with the increase of doping amount.(2)The increase in the dielectric constant of the PI film after nano-SiO_2_ hybridization is due to the larger dielectric constant of the nano-SiO_2_ particles, and the combination with the PI matrix increases the polarization mode. The addition of nanoparticles shortens the electronic transition distance of space charge, increases the generation of probabilistic space charge, and leads to an increase in dielectric loss and conductivity.(3)The corona-resistant lifespan of PI/nano-SiO_2_ composite films increases significantly with the increase of nano-SiO_2_ doping, combined with the electric aging threshold. When the applied electric field strength is extrapolated to 20 kV/mm, the corona resistance time of 10 wt% PI/SiO_2_ three-layer composite film is 5166.75 h. When the applied electric field strength is extrapolated to 20 kV/mm, the corona resistance time of 15 wt% PI/SiO_2_ three-layer composite film is 9518.57 h.

## Figures and Tables

**Figure 1 polymers-14-05469-f001:**
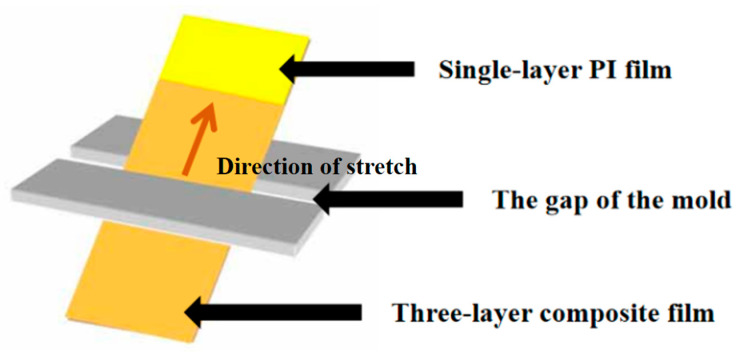
Schematic diagram of the procedure used to prepare the composite films.

**Figure 2 polymers-14-05469-f002:**
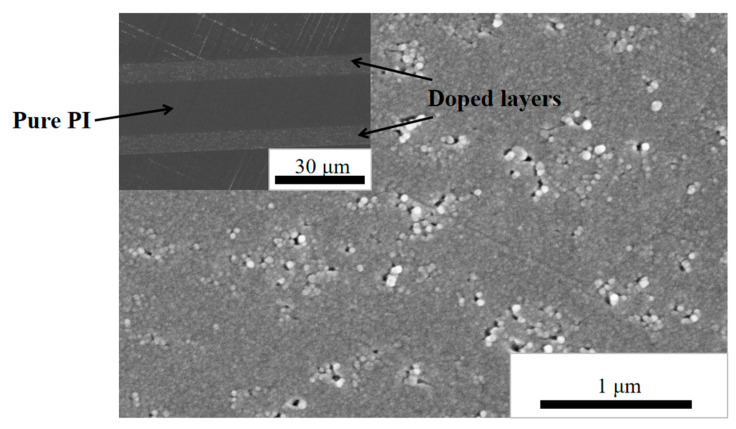
SEM cross-sectional morphologies of PI/nano-SiO_2_ three-layer composite films doped with 5 wt%.

**Figure 3 polymers-14-05469-f003:**
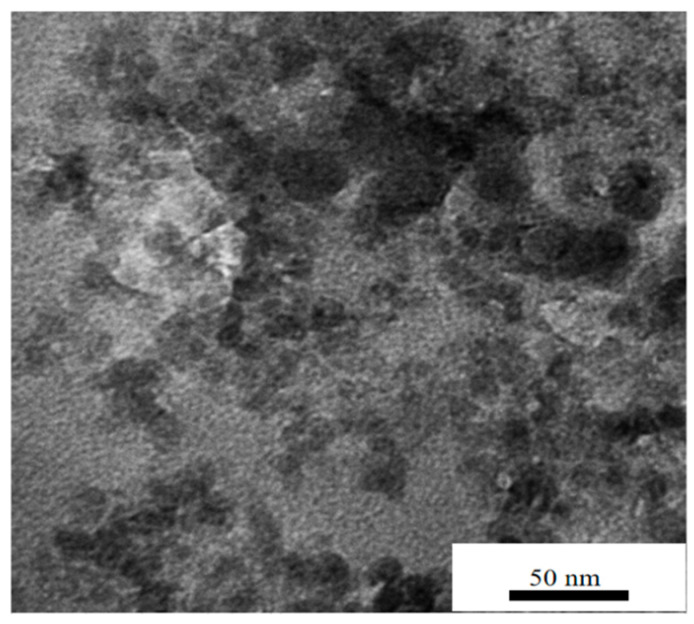
TEM morphologies of PI/nano-SiO_2_ three-layer composite films doped with 5 wt%.

**Figure 4 polymers-14-05469-f004:**
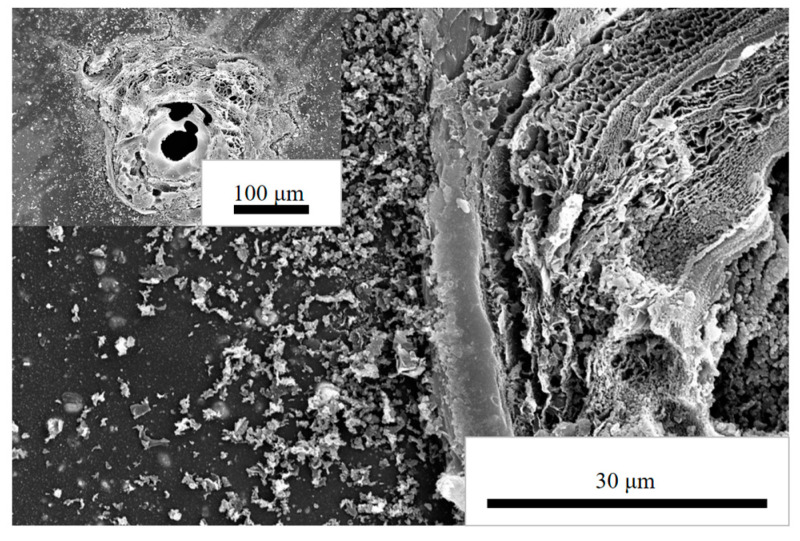
SEM surface morphology near the corona breakdown point of PI/nano-SiO_2_ composite film with doping amount of 5 wt%.

**Figure 5 polymers-14-05469-f005:**
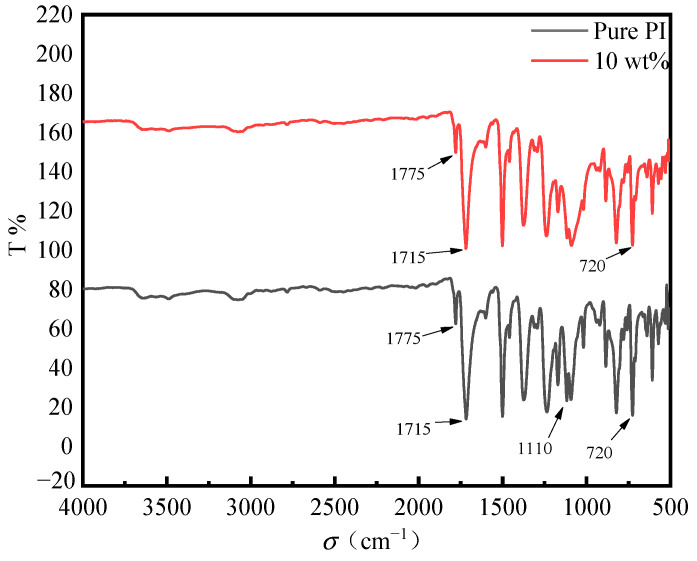
Infrared spectra of pure PI film and PI/nano-SiO_2_ three-layer composite film with doping amount of 10 wt%.

**Figure 6 polymers-14-05469-f006:**
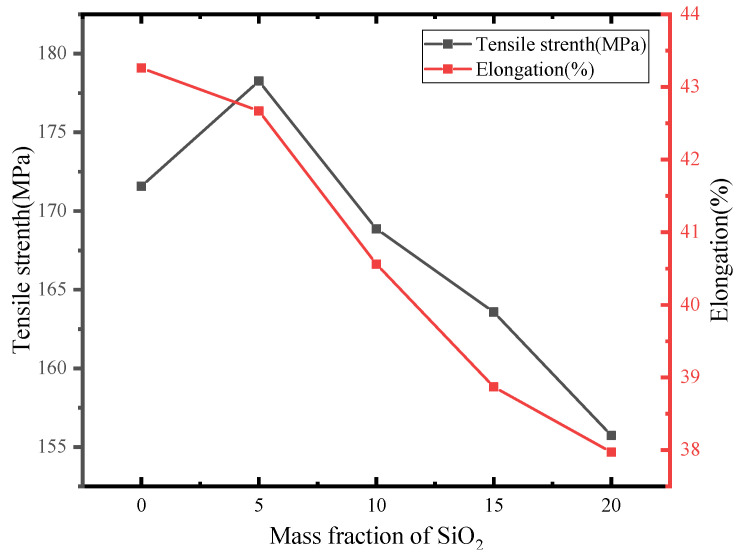
Tensile strength and elongation of pure PI film and PI/nano-SiO_2_ three-layer composite film with different doping amounts.

**Figure 7 polymers-14-05469-f007:**
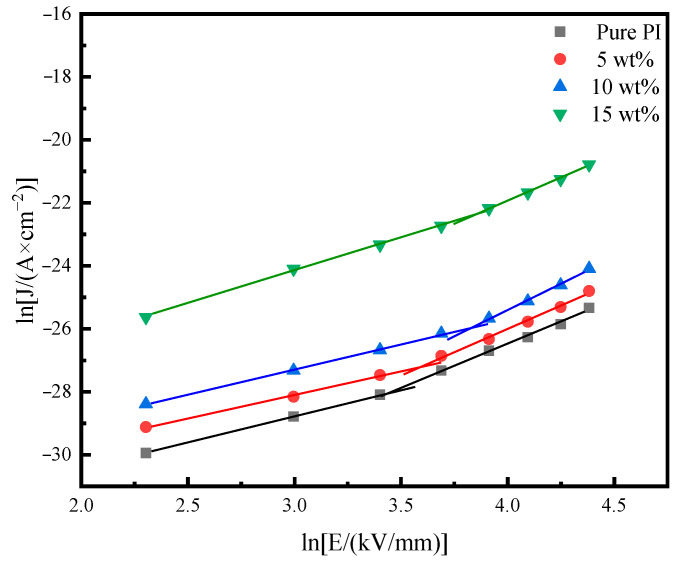
Conductivity and current characteristics of pure film and PI/SiO_2_ composite films with different doping amounts at 155 °C.

**Figure 8 polymers-14-05469-f008:**
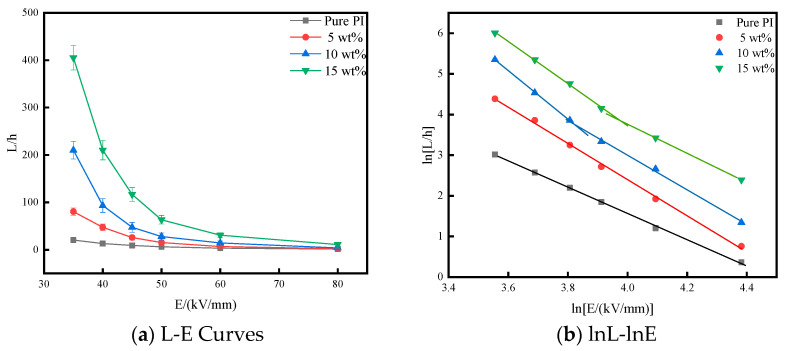
Relationship between field strength and corona-resistant lifespan of pure PI film and PI/SiO_2_ composite film at 155 °C, (**a**) is the field strength and corona-resistant lifespan curves, (**b**) is logarithm of electric field strength and corona-resistant lifespan.

**Figure 9 polymers-14-05469-f009:**
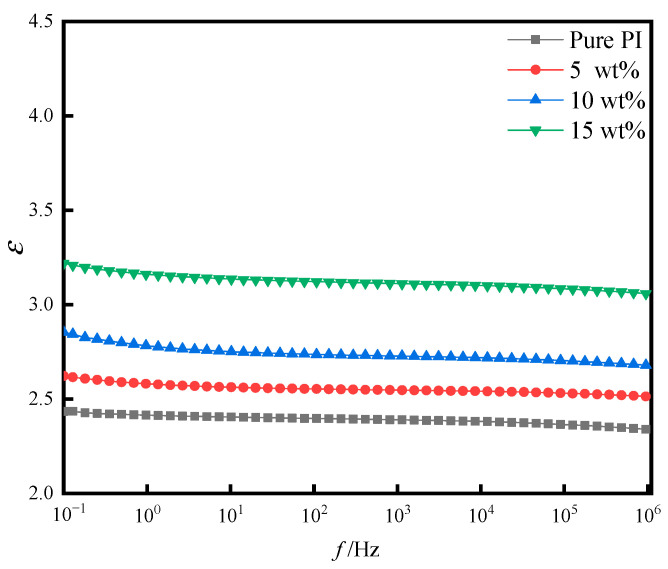
Dielectric constant of pure PI film and different doping amounts of PI/SiO_2_ composite films with frequency.

**Figure 10 polymers-14-05469-f010:**
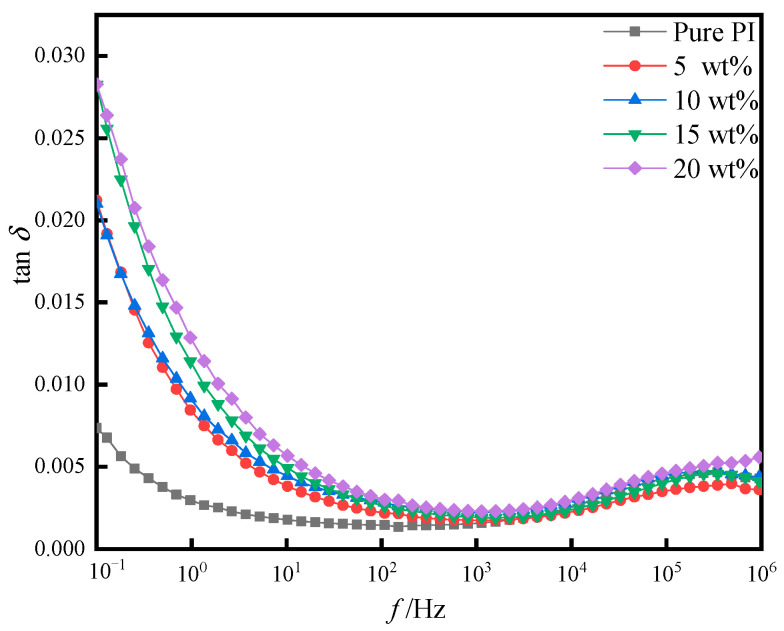
Dielectric loss of pure PI film and different doping amounts of PI/SiO_2_ composite films with frequency.

**Figure 11 polymers-14-05469-f011:**
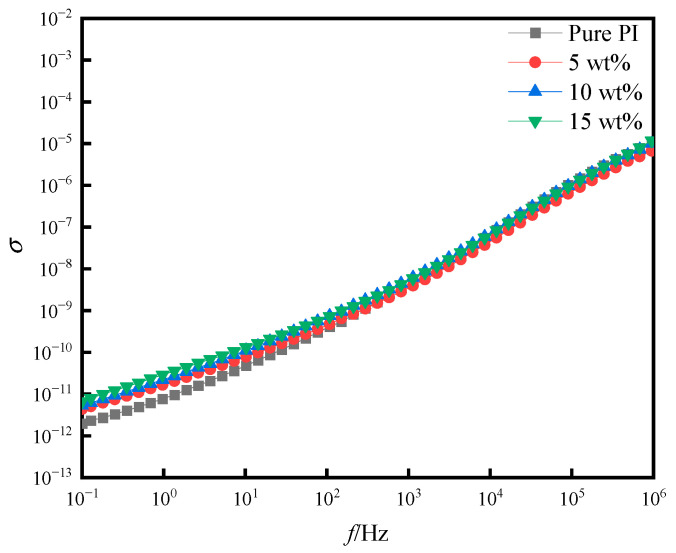
Conductivity of pure PI film and different doping amounts of PI/SiO_2_ composite films with frequency.

**Table 1 polymers-14-05469-t001:** Electrical aging thresholds of pure PI films and PI/SiO_2_ three-layer composite films with different doping amounts.

Sample	Pure PI	5 wt%	10 wt%	15 wt%
Electrical aging threshold/(kV/mm)	30.79	32.78	46.04	53.02

**Table 2 polymers-14-05469-t002:** Corona-resistant lifespan of pure PI films and PI/SiO_2_ composite films with different doping amounts at 20 kV/mm.

Sample	Pure PI	5 wt%	10 wt%	15 wt%
Corona resistance time (h)	112.97 *	962.95 *	5790.65	7472.61

* Low confidence for estimating the corona-resistant lifespan of 20 kV/mm.

## Data Availability

Not applicable.

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
