# Peer review of "Corona Resistance Mechanism of Nano-Modified Polyimide"

_polymers, 2022, doi:10.3390/polym14245469_

Round 1
Reviewer 1 Report
1. What is the average size of the nano-SiO2 particles? What is the purpose of aging the nano-SiO2 sol for one week, to retain a stable size and shape? The nano-SiO2 doped PAA glue underwent filtration to remove impurities. Herein, what are impurities, oversized SiO2 particles?
2. How are the PI middle layer and nano-SiO2 doped PI top and bottom layer laminated together, during molding from the gaps or after forming films? Does the temperature profile, 160 °C for 2 min and then 380 °C, create voids in the film due to the fast evaporation DMAc and H2O? If there are voids in the film, will the void decrease the overall dielectric constant?
3. Figure 1 shows the cross-sectional morphology of the three-layer composite film. Is this morphology from pure PI layer or nano-SiO2 doped PI layers? What are the bright domains, SiO2 or voids?
4. Are there air gaps between the nano-SiO2 particle and PI matrices? High-dielectric-constant oxide-doped polymers are constantly plagued by the air gaps between oxide particles and polymer matrices. When the oxide-doped polymers are cooled down from high manufacturing temperatures, e.g., 380 °C in this work, the different coefficients of thermal expansion of oxide particles and polymers will result in air gaps in between. Furthermore, the air gap drastically lowers the overall dielectric constant. The air gap, if any, can be observed using high-resolution SEM. Or its presence is determined by the discrepancy between the actual and calculated dielectric constants based on volume-weighted compositions.
5. What is the role of the pure PI middle layer? If the nano-SiO2 doped PI forms a film of 25 µm, will it show an even higher dielectric constant and better corona resistance? However, does the nano-SiO2 doped PI show lower mechanical performance than pure PI?

Reviewer 2 Report
In this paper, by combination of microemulsifica- 9
tion and supercritical nano-SiO2 dispersions were prepared. self- 10
made mold was used to prepare Polyimide (PI)/nano-SiO2 three-layer composite films. The paper has some interesting results that could make it publishable in the Journal after the following major revisions:
1-The language of the paper in some parts require revision. Please check the English language of the paper.
2-The first two sentences of the abstract should be combined or shortened. It is better to directly go to the main point here and leave any known introductory info in the introduction section of the paper.
3-Define in the abstract what parameters were investigated, what kind of tests were employed and briefly mention the results of such tests.
4-Introduciton should be strengthened. To modify this section the following documents can be consulted:
- Corrosion behavior of TiZrHfBeCu(Ni) high-entropy bulk metallic glasses in 3.5 wt. % NaCl. npj Mater Degrad 6, 77 (2022). https://doi.org/10.1038/s41529-022-00287-5
- Influence of CeO2 addition on forming quality and microstructure of TiCx-reinforced CrTi4-based laser cladding composite coating. Materials characterization, 171. doi: 10.1016/j.matchar.2020.110732
-Microstructural analysis and optical properties evaluation of sol-gel heterostructured NiO-TiO2 film used for solar panels, Ceramics International 45 (3), 3250-3255, 2019.
5-Figure 1 has some scratches. It is better to be replaced by a higher quality figure.
6-Can the authors provide error bars for figure 4? Same goes for figures 5 and 7
7-there is no figure 6. Figure numbering jumps from 5 to 7.
8-Conclusions should be in bullet points.
9-Provide standard deviation for the data point.
Round 2
Reviewer 2 Report
The paper can now be accepted in its present format.